

# Towards sex identification of Asian Palmyra palm (*Borassus flabellifer* L.) by DNA fingerprinting, suppression subtractive hybridization and *de novo* transcriptome sequencing

Kwanjai Pipatchartlearnwong[1], Piyada Juntawong[1,2,3], Passorn Wonnapinij[1,2,3], Somsak Apisitwanich[1,2] and Supachai Vuttipongchaikij[1,2,3]

[1] Department of Genetics, Faculty of Science, Kasetsart University, Bangkok, Thailand
[2] Center of Advanced studies for Tropical Natural Resources, Kasetsart University, Bangkok, Thailand
[3] Omics Center for Agriculture, Bioresources, Food and Health, Kasetsart University (OmiKU), Bangkok, Thailand

Corresponding author
Supachai Vuttipongchaikij,
fsciscv@ku.ac.th

## ABSTRACT

**Background**. Asian Palmyra palm, the source of palm-sugar, is dioecious with a long juvenile period requiring at least 12 years to reach its maturity. To date, there is no reliable molecular marker for identifying sexes before the first bloom, limiting crop designs and utilization. We aimed to identify sex-linked markers for this palm using PCR-based DNA fingerprinting, suppression subtractive hybridization (SSH) and transcriptome sequencing.

**Methods**. DNA fingerprints were generated between males and females based on RAPD, AFLP, SCoT, modified SCoT, ILP, and SSR techniques. Large-scale cloning and screening of SSH libraries and *de novo* transcriptome sequencing of male and female cDNA from inflorescences were performed to identify sex-specific genes for developing sex-linked markers.

**Results**. Through extensive screening and re-testing of the DNA fingerprints (up to 1,204 primer pairs) and transcripts from SSH (>10,000 clones) and transcriptome data, however, no sex-linked marker was identified. Although *de novo* transcriptome sequencing of male and female inflorescences provided ~32 million reads and 187,083 assembled transcripts, PCR analysis of selected sex-highly represented transcripts did not yield any sex-linked marker. This result may suggest the complexity and small sex-determining region of the Asian Palmyra palm. To this end, we provide the first global transcripts of male and female inflorescences of Asian Palmyra palm. Interestingly, sequence annotation revealed a large proportion of transcripts related to sucrose metabolism, which corresponds to the sucrose-rich sap produced in the inflorescences, and these transcripts will be useful for further understanding of sucrose production in sugar crop plants. Provided lists of sex-specific and differential-expressed transcripts would be beneficial to the further study of sexual development and sex-linked markers in palms and related species.

## INTRODUCTION

Asian Palmyra palm (*Borassus flabellifer* L, $2n = 36$) is a dioecious and slow-growing perennial tree, requiring 12–15 years to reach its maturity and produce the first inflorescence (*Kovoor, 1983*). Once flowered, it vigorously and continuously produces flowers and fruits through its lifespan. This palm is found widespread in South and Southeast Asia and provides essential food and economic values throughout its parts including inflorescence sap for producing palm sugar and alcoholic beverages, fruits for consumption and the tree trunk for construction (*Morton, 1988*; *Lim, 2012*). Both sexes are morphologically identical except for the male and female inflorescences, and there are no reliable means of sex identification before the first bloom. Because of the long juvenile period and a preference for female plants for fruit production, growers hesitate to expand the plantation, and this limits the utilization of this palm. Being able to identify the sexes of Asian Palmyra palm at seedling stages will provide better designs for the crop production through optimal male and female ratios, breeding programs, conservation and utilization (*Davis & Johnson, 1987*; *Barfod et al., 2015*; *Sirajuddin et al., 2016*). Molecular markers for sex identification in Asian Palmyra palm is needed.

Asian Palmyra palm belongs to family Arecaceae, which includes many palm species that are among the world commercially important crops, including oil palm (*Elaeis oleifera*), coconut (*Cocos nucifera*) and date palm (*Phoenix dactylifera*) (*Beck & Balick, 1990*; *Dransfield et al., 2005*). Historically, it was placed among these valued crops as for producing palm sugar, alcohol products and its nutritious fruits dating back at least 2,500 years (*Fox, 1977*; *Ferguson, 1888*), before sugarcane gradually replaced its status. Historical and genetic studies suggest that Asian Palmyra palm was originated from tropical Africa, brought along the spice route to the east and settled in the Indian subcontinent, where it is widely propagated throughout India and Sri Lanka, before introduced to Southeast Asia more than 1,500 years ago (*Pipatchartlearnwong et al., 2017a*). Given its extended benefits since the historical time and because of its vigorous growth and continuous supply of inflorescence sap and fruits with the ability to withstand severe climate, arid conditions, pests and diseases, Asian Palmyra palm should be recognized as a potential recalcitrant food crop to the climate change for the tropical and subtropical regions.

Molecular data of Asian Palmyra palm are currently limited. The genome sequence is not available, and only its chloroplast genome was recently reported (*Sakulsathaporn et al., 2017*). A few sex-linked markers have been developed based on DNA fingerprinting, but only one RAPD-based marker was shown to be able to identify the sexes among the populations in India (*George et al., 2007*). However, our preliminary experiment showed that this marker was unable to identify the sexes in the population in Thailand. Previously, we showed that Asian Palmyra palm populations in Thailand are descended from a small number of seedlings brought in at least 1,500 years ago and represent a very narrow genetic diversity (*Pipatchartlearnwong et al., 2017a*; *Pipatchartlearnwong et al., 2017b*).

In this work, we aimed to develop sex-linked markers of Asian Palmyra palm through three approaches: DNA fingerprinting, direct cloning of subtraction subtractive hybridization (SSH) of cDNA from male and female inflorescences and *de novo*

transcriptome sequencing of male and female inflorescences. Extensive sets of PCR primers belonging to different DNA fingerprinting techniques including RAPD, AFLP, SCoT, ILP, TEs, EST-SSR and gSSR and those that specific to transcripts identified from SSH and transcriptome analysis were exhaustively tested to identify sex-linked markers. Although none of the sex-linked markers was obtained from this study, we have narrowed the path towards the sex identification of Asian Palmyra palm. Because identifying sex-linked markers for this species appeared to be extremely difficult as opposed to works in other dioecious plants (*Heikrujam et al., 2015*), we discussed the nature of this work in conjunction with others successfully identified markers with future directions. Furthermore, this work provided the first *de novo* transcriptome sequencing of Asian Palmyra palm. Lists of candidate transcripts that are specific to sexes and developmental stages of male and female inflorescences are presented, and these will be useful for further study on sex determination, sexual development and floral development of Asian Palmyra palm.

## MATERIALS & METHODS

### Plant materials

For DNA isolation, young leaves were collected from mature palm plants with known sexes from various locations in three regions of Thailand: the southern region (Song-Khla and Surat-Thani provinces), the central region (Phachinburi, Phetchaburi, Pathum-Thani, Nakon-Pathom, Kanchanaburi, Nakhon-Sawan and Chainat provinces) and the northeastern region (Nakhon-Ratchasima, Burirum, Ubon-Ratchathani, Kalasin and Amnat-Charoen provinces). For RNA isolation, male and female inflorescences were collected from the southern region (Song Khla province) and the central region (Nakon-Pathom province).

### Nucleic acid isolation

Total DNA was isolated from young leaves using a modified method based on the CTAB method as described previously in *Pipatchartlearnwong et al. (2017a)* and *Pipatchartlearnwong et al. (2017b)*. Total RNA was isolated from young inflorescences (see Fig. 1) using the modified CTAB method. Briefly, the sample was pulverized into fine powder in liquid $N_2$ by using a mortar and pestle and mixed with CTAB extraction buffer [2% (w/v) CTAB, 100 mM Tris-HCl pH.8, 20 mM EDTA and 1.4 M NaCl, 2% (w/v) polyvinylpyrrolidone-90 and 2% (v/v) β-mercaptoethanol]. RNA was then precipitated using 1/3 volume of 10 M LiCl at $-20\,°C$ for overnight and then centrifuged at $11,750\times$ g at $4\,°C$ for 30 min. Total RNA was treated with DNase I (New England BioLabs®Inc., Ipswich, MA, USA) at 37 °C for 30 min followed by phenol: chloroform extraction and ethanol precipitation. RNA quality and quantity were analyzed by agarose gel electrophoresis and NanoDrop (Thermo Scientific, Waltham, MA, USA).

### DNA fingerprinting

The RAPD-based OPA-06 marker was conducted according to *George et al. (2007)* using total DNA from 20 male and 20 female samples. AFLP analysis was performed using 26

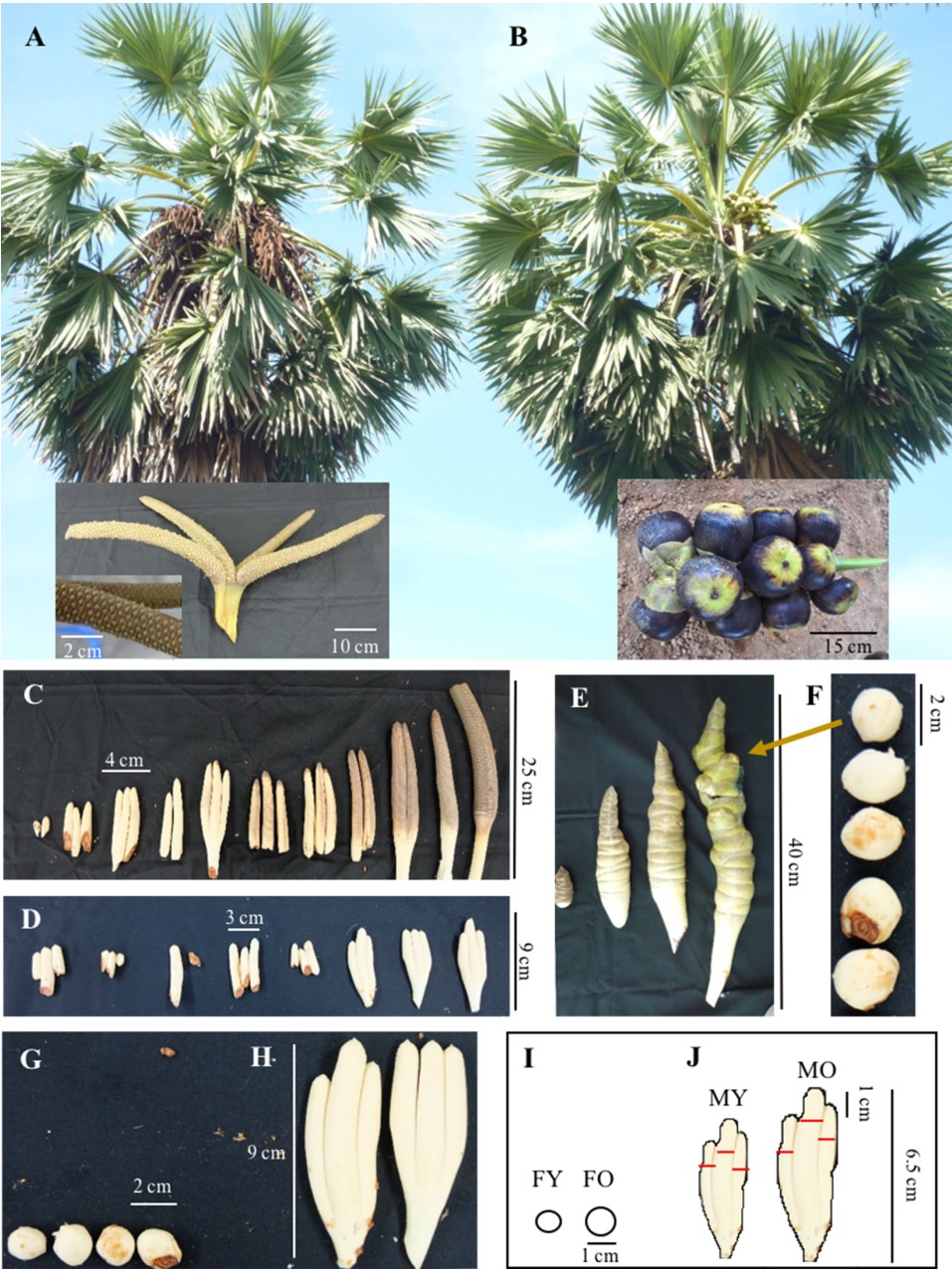

**Figure 1  Asian Palmyra palm samples from male and female plants.** (A) A mature male plant with a fully developed male inflorescence and (B) a mature female plant with a female inflorescence bearing fruits. (C) and (D) Representatives of developing male inflorescences from young to mature stages. (E) Representatives of developing female inflorescences, and (F) buds extracted from female inflorescences. (G–J) Representative female and male inflorescence samples used for the SSH experiment (G and H, respectively) and *de novo* transcriptome sequencing analysis (I and J, respectively). Scale bars are indicated within.

male and 46 female samples by, firstly, restriction digestion of 250 ng total DNA using EcoRI and MseI (Thermo Fisher Scientific, USA) and adaptor ligation. And, secondly, pre-selective amplification was performed in a 25 µl reaction volume containing 2 µl of the digested DNA, 5 mM dNTPs, 40 mM MgCl2, 5 ρM of each MseI adaptor+C and EcoRI adaptor+A primers and 1 unit of *Taq* DNA polymerase (Vivantis, Malaysia) using the conditions as follows: 20 cycles of 94 °C for 30 s, 56 °C for 1 min and 72 °C for 1 min, with a final extension step at 72 °C of 5 min. The reaction was diluted 20-fold using dH$_2$O before being used in a selective amplification reaction: 25 µl total volume containing 5 µl of the diluted DNA, 4 mM dNTPs, 40 mM MgCl2, 5 ρM MseI+3 and 5 ρM of EcoRI+3 primers (Table S1) and 1 unit of *Taq* DNA polymerase using a touch-down condition (12 cycles of 94 °C for 30 s, 65 °C (−0.7 °C/cycle) for 30 s and 72 °C for 1 min, followed by 23 cycles using the annealing temperature at 56 °C). SCoT was performed using cDNA from four male and four female samples. SCoT primers were based on *Collard & Mackill (2009)* (Table S2). The PCR reaction was performed in a 20 µl volume, which included 50 ng of DNA, 4.8 mM dNTPs, 30 mM MgCl2, 20 ρM SCoT primer and 1 unit of *Taq* polymerase (Vivantis, Malaysia) using 35 cycles of 94 °C for 1 min, 50 °C for 1 min and 72 °C for 2 min, with a final extension for 5 min. The modified SCoT method was performed by mixing SCoT primers with primers from polyA, EST-SSRs of oil palm and transposon element (TE) markers (Tables S3 and S4) (*Bureau & Wessler, 1994*; *Takata, Kishima & Sano, 2005*). ILP marker based on *Ukoskit & U-thoomporn (2016)* (Table S5) was performed using total DNA from three male and three female samples. Each PCR reaction were performed in a 20 µl volume containing 20 ng of DNA, 4 mM dNTPs, 75 mM MgCl2, 3.5 ρM for each primer and 0.5 unit of *Taq* polymerase (Vivantis, Selangor, Malaysia) using 35 cycles of 94 °C for 30 s, 56 °C for 1.30 min and 72 °C for 30 s, with a final extension for 5 min. EST-SSR and gSSR microsatellite markers (Table S6 and S7, based on *Arabnezhad et al., 2012*; *Billotte et al., 2004*; *Elmeer et al., 2011*; *Pipatchartlearnwong et al., 2017a*; *Pipatchartlearnwong et al., 2017b*) were performed using total DNA from four males and four female samples using the same PCR condition as above. PCR products for RAPD, SCoT and ILP were resolved in 2% (w/v) agarose gel electrophoresis and visualized under ultraviolet (UV) light after staining with ethidium bromide. Those for AFLP, SCoT+EST-SSR, SCoT+TEs, EST-SSR and gSSR were resolved in 6% (w/v) polyacrylamide gel electrophoresis and visualized by silver staining.

## Suppression subtractive hybridization (SSH)

Total RNA from one male and one female inflorescence samples from Song Khla province were used in the study (Figs. 1G–1H). The first-strand and double strand cDNA was synthesized from 1 µg of total RNA samples using the SMART cDNA Library Construction Kit (Clontech Laboratories Inc., Palo Alto, CA, USA) following to the manufacturer protocol. Double-strand cDNA of male and female was labeled with biotin-16 dUTP using nick translation: 100 µl total volume containing 400 ng of double-strand cDNA, 3.96 mM of dGTP, dATP and dCTP, 1.6 mM of dTTP, 2.4 µl of biotin-16 dUTP (Sigma-Aldrich), 48 µM of 5′PCR PrimerII A (5′-AGCAGTGGTATACAACGCAGAGT-3′), 10x *Taq* buffer, and 5 unit of *Taq* polymerase (Vivantis, Selangor, Malaysia) using 40 cycles of 94 °C for

15 s, 65 °C for 30 s, and 68 °C for 6 min. The biotin-16 dUTP labeled cDNA was checked for its detection signal using dot blot hybridization at least at $10^{-4}$ dilution. First-strand cDNA of male and female cDNA was subtracted using labeled double-strand cDNA of its opposite sex at ratio 1:3 following a method from *Rebrikov et al. (2004)*. Subtracted samples were purified by ethanol precipitation, and double-strand cDNA was synthesized by PCR reaction containing 2 μl of subtracted cDNA, 48 μM of 5′ PCR PrimerII A, 10 mM dNTPs, 5x NEB polymerase buffer and 5 unit of Q5® High-Fidelity DNA polymerase (New England Biolabs) through 21 cycles of 94 °C for 15 s, 65 °C for 30s and 68 °C for 6 min. Subtracted double-strand cDNA was purified and ligated into pGEM-Teasy (Promega). Clones were selected by blue-white colony section and screened using colony PCR for > 500 bp inserted fragment, before re-selection using dot blot hybridization I and II using probes from opposite sexes. Selected clones were sequenced and analyzed against GenBank using BLAST.

## Transcriptome analysis

Transcriptome sequencing was performed by Macrogen Inc. (Republic of Korea) using 10 μg of total RNA. Total RNA was obtained from two male (MY and MO) and two female (FY and FO) inflorescence samples from Nakon-Pathom province (Figs. 1I–1J). Briefly, cDNA libraries were constructed using the TruSeq™ RNA sample preparation kit (Illumina, USA) and sequenced on a HiSeq 2000 (Illumina, USA) with paired-end 100 bp read lengths. Initial raw reads were trimmed and filtered with FastQC (http://www.bioinformatics.babraham.ac.uk/projects/fastqc) and Trimmomatic version 0.32 (http://www.usadellab.org/cms/?page=trimmomatic) software to eliminated low-quality reads (quality score lower than 20) and remove adapters. Reads were considered as high quality if more than 70% of the bases had Phred values more than Q20. Reads produced from this study were assembled by Trinity software (version r20140717) using default parameters (*Grabherr et al., 2011*). For similarity search, the assembled transcripts were blasted against the NCBI non-redundant protein sequence database and TAIR database using Blast2GO with the e-value cutoff <10-10. RSEM version 1.2.15 software was used to estimate transcript abundance (*Li & Dewey, 2011*). The assembled sequences were analyzed for Gene Ontology (GO) and Kyoto Encyclopedia of Genes and Genomes (KEGG: *Kanehisa & Goto, 2000*) using Blast2GO. For differential gene expression analysis, Fastq files were aligned to the assembled transcriptome using TopHat2 alignment program (*Kim et al., 2013*). Transcriptome annotation file (GFF) was built using the Cufflinks program (*Trapnell et al., 2010*) by performing a combined assembly of four transcriptome datasets (FY, FO, MY, and MO). Transcript candidates for male and female were chosen using parameters as follows; male-highly represented transcripts [FPKM > 10 and > 0.5 for MY or MO and FPKM = 0 for both FY and FO] and [FPKM > 5 for both MY and MO and FPKM = 0 for both FY and FO] with length > 300 bp, and female-highly represented transcripts [FPKM > 10 and > 0.5 for FY or FO and FPKM = 0 for both MY and MO] and [FPKM > 5 for both FY and FO and FPKM = 0 for both MY and MO] with length > 300 bp. To identify shared transcripts among the four datasets, transcripts from each dataset with FPKM > 5 and length > 300 bp were analyzed using Venny version 2.1.0. Identified male

**Table 1  Total markers tested for DNA fingerprints between male and female plants.** Numbers of male and female samples used for each marker are indicated in brackets as M and F, respectively.

| Marker | Total tested markers | Amplifiable | Polymorphic marker (loci) | Sex polymorphic loci | Sex-linked marker |
|---|---|---|---|---|---|
| RAPD | 1 | 1 (F23:M20) | 0 | 0 | – |
| AFLP | 36 | 36 (F46:M26) | 13 (141 loci) | 0 | – |
| SCoTs | 36 | 36 (F4:M4) | 3 | 0 | – |
| SCoTs/A | 36 | 36 (F4:M4) | 3 | 0 | – |
| SCoTs (36)/EST-SSRs (3) | 108 | 64 (F4:M4) | 7 (7 loci) | 2 (F8:M8) | – |
| SCoTs (36)/TEs (11) | 396 | 163 (F4:M4) | 21 (48 loci) | 48 (F4:M4) | – |
| Oil palm ILPs | 41 | 36 (F3:M3) | 0 | 0 | – |
| Date palm gSSRs (high PIC values) | 5 | 2 (F4:M4) | 0 | 0 | – |
| Oil palm EST-SSRs | 289 | 150 (F20:M20) | 11 (17 loci) | 0 | – |
| Oil palm gSSRs | 256 | 168 (F20:M20) | 8 (12 loci) | 0 | – |
| Total | 1,204 | 583 | 66 (231) | 50 | – |

and female specific transcripts were subjected to GO enrichment analysis using Fisher's Exact Test with FDR cutoff = 0.01. Differentially expressed genes (DEGs) among the four transcript datasets were analyzed by Cuffdiff with FDR cutoff <0.05.

# RESULTS

## Extensive DNA fingerprinting analysis failed to identify any sex-linked marker for Asian Palmyra palm in Thailand

Previously, *George et al. (2007)* had developed a RAPD based male-specific marker (OPA-$06_{600}$) for the Asian Palmyra palm population in India. Initially, we tested this marker for sex identification in our population in Thailand using up to 20 male and 24 female individuals, but this marker was unable to confirm the sexes (Fig. S1). To identify sex-linked markers for Asian Palmyra palm in Thailand, we generated male and female DNA fingerprints using ten different DNA fingerprinting techniques as presented in Table 1. Although a number of potential sex-linked bands were obtained from SCoTs/EST-SSRs and SCoTs/TEs, after sequencing and re-testing these bands using specific primers, no sex-linked marker was obtained. Despite such extensive screening of DNA markers up to 1,204 primer pairs by the ten techniques, we did not obtain any sex-linked marker. This experiment showed that the DNA fingerprinting covered here is inadequate for identifying a sex-linked marker for Asian Palmyra palm. It also suggests that sex determination region in this species could be of small and very difficult to identify. Further attempts for this scheme in the Asian Palmyra palm should be aware of this limitation. Other means of sex identification for Asian Palmyra palm should be explored.

## Identification of sex-related transcripts by SSH analysis using male and female inflorescence flowers

To identify genes related to sexes, we performed SSH using total RNA isolated from male and female inflorescences of Asian Palmyra palm. Because the floral development of this species is not well defined, we collected young inflorescence stems from male and female

**Table 2  Clone selection of SSH between male and female.**

| Library | Female | Male |
|---|---|---|
| Direct cloning by blue-white colony selection | 9,820 | 2,574 |
| Fragment size > 500 bp by PCR screening | 4,097 | 1,288 |
| Dot blot hybridization I | 498 | 112 |
| Dot blot hybridization II | 63 | 81 |
| BlastX | 29 | 60 |

plants as soon as they emerged from the dense leaf sheets and isolated young floral tissues for RNA isolation. Female flower buds (∼2 cm in diameter) were removed from the inflorescence (Fig. 1G), while the male inflorescences (∼9 cm in length) were used as a whole (Fig. 1H). Direct cloning of subtracted-cDNA yielded 2,574 and 9,820 clones for male and female, respectively, and, after colony-PCR screening for > 500 bp inserted-fragments, we obtained 1,288 and 4,097 clones for male and female libraries, respectively (Table 2). These selected clones were re-tested against their opposite sex using two rounds of dot blot hybridization (I and II), resulting in 81 and 63 clones for male- and female-specific libraries, respectively. These clones were subsequently sequenced and searched in GenBank using BLASTX, and 60 male and 29 female clones were found matching to non-redundant genes in the plant database (Table S8). These sequences have been deposited in GenBank (JZ977504–JZ977592) as ESTs for male or female inflorescence flowers of Asian Palmyra palm.

Among the total 99 identified clones, 91 and 98 clones share high similarities to sequences in the nuclear genome of date palm and oil palm, respectively. As oil palm has both male and female genome sequences available, we observed that all 99 sequences identified in Asian Palmyra palm are present in both male and female genomes of oil palm, suggesting that these sequences could not be used as sex-specific markers. Nonetheless, we tested 19 selected sequences on gDNA from male and female plants by PCR using specific primers (Table S9), and these failed to identify the sexes. Although this experiment was unable to provide a sex-linked marker, the list of expressed genes during male- and female-floral development of Asian Palmyra palm could be of use for future study.

### De novo transcriptome sequencing of male and female inflorescences

To further identify sex-related genes for sex identification, we performed *de novo* transcriptome sequencing using RNA from male and female inflorescences. Four cDNA libraries were constructed from two floral stages of male and female (FY-female young inflorescences, FO-female old inflorescences, MY-male young inflorescences and MO-male old inflorescences; Figs. 1I–1J), which were in earlier stages than those used in the SSH experiment. From Illumina HiSeq2000 sequencing, we obtained 47,194,682-90,305,610 reads for each sample with an average length of 98 bp, after trimming the adapter sequences and removing low-quality nucleotides (<Q20) and short sequences (<25 nt) (Fig. 2). Sequence assembly using Trinity yielded 187,083 transcripts with 705 bp average length. Most of the assembled transcripts were between 200 and 300 bp in length, and up to 70,785 transcripts were > 500 bp in length (Fig. 2B). The transcripts were

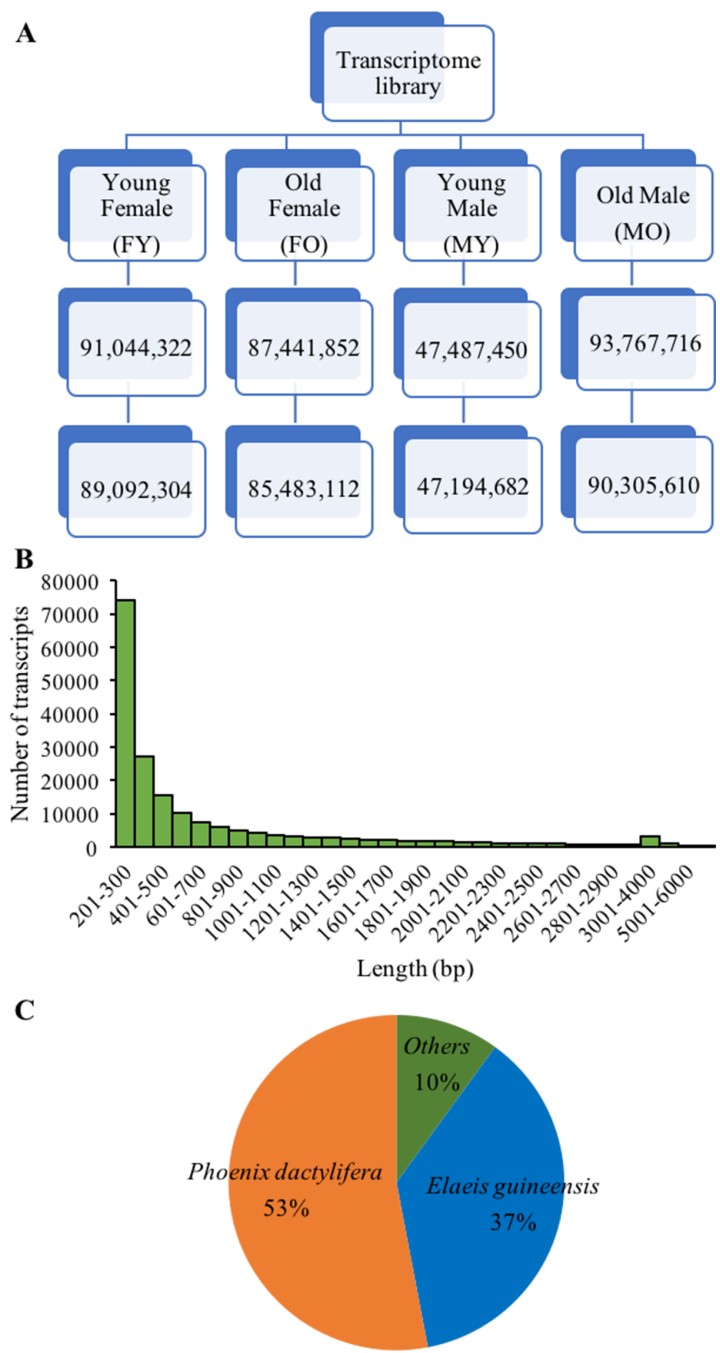

**Figure 2** *De novo* **transcriptome sequencing overview.** (A) Total sequence reads (bp) after trimming and removing low quality read (<Q20) of four samples. (B) Distribution of number and length of assembled transcripts, and (C) sequence similarity of assembled transcripts to other plant species.

annotated by using BLASTX searches against the NCBI non-redundant protein database, and 77,578 transcripts (41.47% of initial transcripts) were identified as unique sequences with significant similarities (E <1e−10) to known protein sequences from 5,331 different

species. The average alignment length (%) matched was 81.3% with the highest positive matched per alignment length being 90%, and the average percent identity was 81.24%. Most of these sequences were identified as similar sequences to those found in date palm (53%) and oil palm (37%) (Fig. 2C). The Transcriptome Shotgun Assembly are deposited at DDBJ/ENA/GenBank under the accession GFYQ00000000.

## GO classification and pathway assignment by KEGG

Gene ontology (GO) terms of the 77,578 annotated transcripts were assigned using Blast2GO program in three categories: biological process, cellular component and molecular function. The transcripts were assigned into 66 functional groups, and top GO terms with more than 1% assigned transcripts in each category are presented in Fig. S2. Binding and catalytic activity were dominant in the molecular function category, while the integral component of the membrane and nucleus dominated the cellular component category. For the biological process, oxidation–reduction process and protein phosphorylation were the most represented groups. Subsequently, function classification and pathway assignments based on Kyoto Encyclopedia of Genes and Genomes (KEGG) showed that, among the 77,578 annotated transcripts, 16,635 were annotated with enzyme code EC numbers and mapped into 139 KEGG pathways. Pathways with more than 1% matched transcripts are presented in Fig. S3. Purine (map00230) and Thymine (map00730) metabolisms were the most matched pathways with 4,913 transcripts (18.89%) and 2,581 transcripts (9.92%), followed by Biosynthesis of antibiotics (map01130) with 1,582 transcripts (6.08%). Interestingly, starch and sucrose metabolism (map00500) presented at the fourth rank with 1,187 transcripts (4.56%). Transcripts mapped into this pathway were mostly related to sugar metabolism for fructose and sucrose production, but less supported to starch biosynthesis (Fig. 3 and Table 3; see the transcript IDs in Table S10). Other related sugar metabolic pathways were also found among the list of top pathways including 1.54% of glycolysis/gluconeogenesis (map00010), 1.25% of pentose and glucuronate interconversions (map00040), 1.19% of Galactose metabolism (map00052), 1.10% of fructose and mannose metabolism (map00051), 0.87% of pentose phosphate pathway (map00030) and 0.85% of inositol phosphate metabolism (map00562). This observation coincides with the facts that both male and female inflorescences of Asian Palmyra palm produce sweet sap, which has been used for making palm sugar for centuries.

## Differential expression of genes between male and female inflorescences

Among the initial 187,083 transcripts, we observed a number of transcripts that were highly represented in either male or female datasets, but none in their opposite sex (FPKM = 0); 33 and 11 transcripts were identified from female and male datasets, respectively (length > 300 bp, FPKM > 5 for both FY and FO or MY and MO, and FPKM > 10 and > 0.5 for FY or FO and MY or MO) (Table S11). Although annotations of these transcripts did not show any evidence related to genes for male- or female-specific development, we observed four cell wall-related transcripts in the female datasets, and this may reflect that the developing female flower undergoes rapid and extensive organ enlargement, whereas the male flower

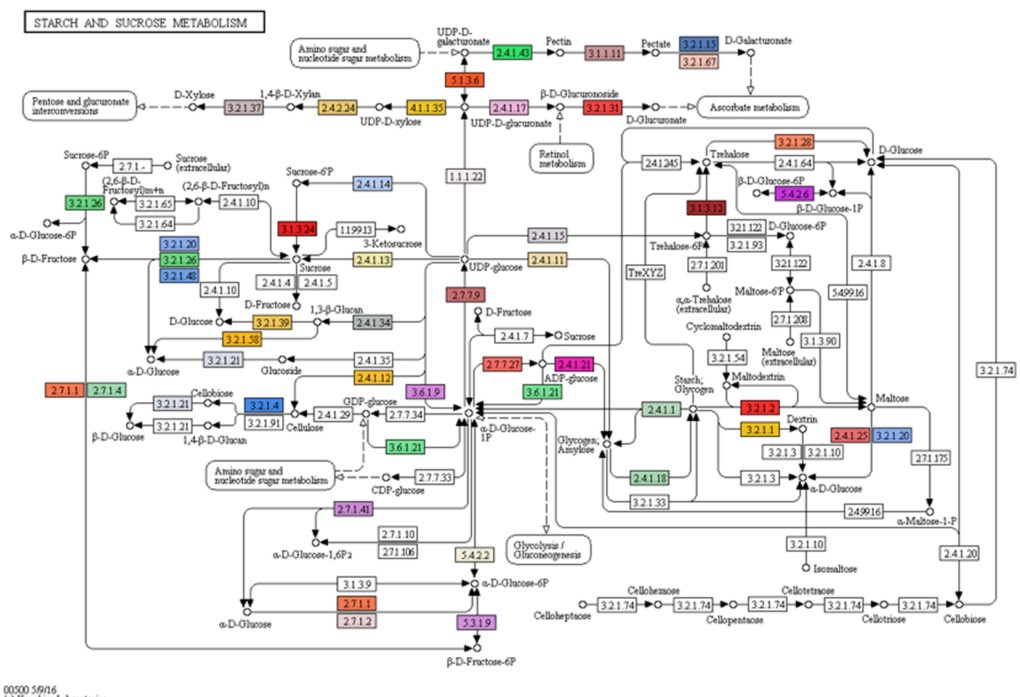

**Figure 3** **Mapping of assembled transcripts onto the starch and sucrose metabolism (map00500) taken from KEGG.** Mapped enzymes are indicated by colors (see Table S10 for details). Kanehisa Laboratories 00500 5/9/16.

is limited to a small size. Because the genome sequence of Asian Palmyra palm is currently unavailable, we thus tested whether these transcripts could be used for sex identification by PCR using male and female gDNA and primers specific to these 44 transcript sequences (Table S12). However, these primers gave similar band patterns between male and female gDNA (Fig. S4), indicating that these primers cannot be used for identifying the sexes.

By comparing the transcripts among the four datasets (length >300 bp and FPKM > 5), we found that 17,231 transcripts were shared among all datasets, and 4,514 (14.4%) and 3,312 (10.5%) transcripts were identified as male- and female-specific transcripts, respectively (Fig. 4 and Table S13). These transcripts were further classified into specific MY (2,192 transcripts), MO (1,405 transcripts), FY (1,355 transcripts) and FO (1,165 transcripts) to identify transcripts that may relate to male or female floral development stages. Subsequently, we analyzed enriched GO terms for the male- and female-specific datasets (length > 300 bp and FPKM > 5), and top enriched GO terms (FDR <1E−02) are presented in Table 4 (see Table S14 for transcript IDs). All GO terms observed here were over-represented compared to the reference sets. Interestingly, carbohydrate metabolism and cell wall-related processes were much apparent in the female-specific transcripts, while various catabolic processes for biological compounds dominated the male-specific transcripts.

In light of this transcriptome analysis, we cross-referenced the 60 male and 29 female-specific clones from the SSH experiment to the transcript abundance data. Noting that

**Table 3  The list of enzymes in starch and sucrose metabolism (KEGG map00500) identified in male and female transcripts from inflorescences of Asian Palmyra palm.**

| Enzyme | EC number | Number of transcripts |
|---|---|---|
| decarboxylase | EC:4.1.1.35 | 11 |
| phosphodismutase | EC:2.7.1.41 | 1 |
| endo-1,4-beta-D-glucanase | EC:3.2.1.4 | 56 |
| saccharogen amylase | EC:3.2.1.2 | 42 |
| glycogenase | EC:3.2.1.1 | 20 |
| 1,4-alpha-galacturonidase | EC:3.2.1.67 | 5 |
| isomerase | EC:5.3.1.9 | 6 |
| alpha-glucosidase | EC:3.2.1.48 | 42 |
| 1,3-beta-glucosidase | EC:3.2.1.58 | 6 |
| adenylyltransferase | EC:2.7.7.27 | 15 |
| maltase | EC:3.2.1.20 | 2 |
| gentiobiase | EC:3.2.1.21 | 106 |
| synthase | EC:2.4.1.34 | 39 |
| trehalase | EC:3.2.1.28 | 5 |
| invertase | EC:3.2.1.26 | 45 |
| beta-glucuronide glucuronohydrolase glucuronidase | EC:3.2.1.31 | 6 |
| endo-1,3-beta-D-glucosidase | EC:3.2.1.39 | 30 |
| 4-alpha-galacturonosyltransferase | EC:2.4.1.43 | 59 |
| 1,4-beta-xylosidase | EC:3.2.1.37 | 29 |
| phosphorylase | EC:2.4.1.1 | 28 |
| branching enzyme | EC:2.4.1.18 | 7 |
| 1-naphthol glucuronyltransferase | EC:2.4.1.17 | 33 |
| synthase (UDP-forming) | EC:2.4.1.15 | 29 |
| synthase | EC:2.4.1.14 | 18 |
| synthase | EC:2.4.1.13 | 44 |
| synthase (UDP-forming) | EC:2.4.1.12 | 98 |
| synthase | EC:2.4.1.11 | 24 |
| disproportionating enzyme | EC:2.4.1.25 | 8 |
| synthase (glycosyl-transferring) | EC:2.4.1.21 | 13 |
| pectin depolymerase | EC:3.2.1.15 | 75 |
| diphosphatase | EC:3.6.1.9 | 9 |
| glucokinase (phosphorylating) | EC:2.7.1.2 | 14 |
| hexokinase type IV glucokinase | EC:2.7.1.1 | 44 |
| fructokinase (phosphorylating) | EC:2.7.1.4 | 35 |
| trehalose 6-phosphatase | EC:3.1.3.12 | 58 |
| 4-epimerase | EC:5.1.3.6 | 4 |
| diphosphatase | EC:3.6.1.21 | 7 |
| phosphatase | EC:3.1.3.24 | 6 |
| uridylyltransferase | EC:2.7.7.9 | 7 |
| pectin demethoxylase | EC:3.1.1.11 | 69 |
| synthase | EC:2.4.2.24 | 6 |
| (alpha-D-glucose-1,6-bisphosphate-dependent) | EC:5.4.2.2 | 9 |
| beta-pgm (gene name) | EC:5.4.2.6 | 3 |
| 6-dehydrogenase | EC:1.1.1.22 | 14 |

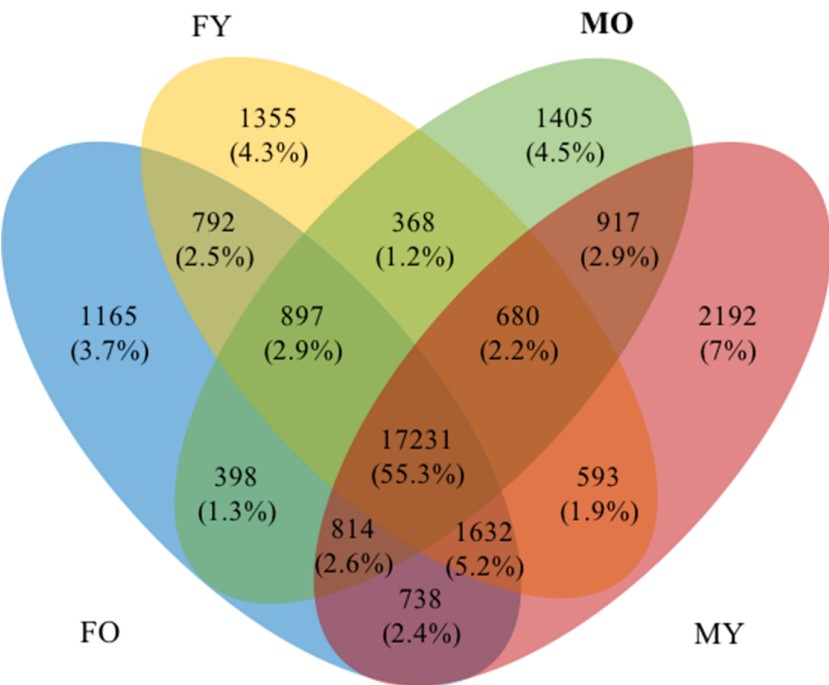

**Figure 4** **A Venn diagram of four transcript datasets.** Numbers and percentages of transcripts are indicated.

RNA samples used in the SSH experiment were obtained from an older inflorescence stage than those used for transcriptome analysis. The transcript IDs were readily identified with more than 90% identical matches and > 300 transcript length (Table S15). However, we found that, based on FPKM values, the sex-specific clones were indeed uncorrelated to almost all of the transcript data, and only F152 (predicted proline-rich protein 4-like) clone could be identified for their expression towards the female. Though, this clone has no direct relationship to sex development based on the annotation.

To further verify sex-related genes, we analyzed differentially expressed genes among the four datasets by using Cuffdiff with FDR <0.05 cutoff. Initially, 816 transcripts were identified, but only 43 transcripts displayed differential expression between sexes with transcript length > 300 bp, FPKM > 5, GO terms and significant $q$ value (<0.05) (Table 5). Among these 43 transcripts, seven transcripts were annotated with genes previously identified to be involved in sex determination and flower development: two and five transcripts for female and male datasets, respectively. Furthermore, we observed six and one transcripts encoding transcription factors that were highly expressed in female and male datasets, respectively. Although being identified for differential expression between the sexes, c1819_g1_i1 and c142400_g1_i1 transcripts encoding ethylene-responsive transcription factors were highly expressed throughout the four datasets, and these genes may be required for the floral development. The list of gene candidates indicated here could be used for a further study on sex determination and floral development in Asian Palmyra palm and other related palm species.

**Table 4  Enriched GO terms of male- and female-specific transcripts (FDR <1E−02).**

| Enriched female | | | Enriched male | | |
|---|---|---|---|---|---|
| **Go term** | **Cat** | **FDR** | **Go term** | **Cat** | **FDR** |
| carbohydrate metabolic process [GO:0005975] | P | 4.98E−05 | L-allo-threonine aldolase activity [GO:0008732] | F | 1.12E−09 |
| phosphoribulokinase activity [GO:0008974] | F | 4.98E−05 | threonine aldolase activity [GO:0043876] | F | 1.12E−09 |
| cell wall organization or biogenesis [GO:0071554] | P | 4.98E−05 | aspartate family amino acid catabolic process [GO:0009068] | P | 1.34E−08 |
| plant-type cell wall organization or biogenesis [GO:0071669] | P | 1.09E−03 | threonine catabolic process [GO:0006567] | P | 1.34E−08 |
| fucose metabolic process [GO:0006004] | P | 1.09E−03 | glycine biosynthetic process [GO:0006545] | P | 3.60E−08 |
| cell wall biogenesis [GO:0042546] | P | 1.92E−03 | isovaleryl-CoA dehydrogenase activity [GO:0008470] | F | 9.54E−06 |
| plant-type secondary cell wall biogenesis [GO:0009834] | P | 2.50E−03 | threonine metabolic process [GO:0018927] | P | 1.20E−05 |
| plant-type cell wall biogenesis [GO:0009832] | P | 2.50E−03 | simple leaf morphogenesis [GO:0060776] | P | 8.86E−05 |
| cellular carbohydrate biosynthetic process [GO:0034637] | P | 2.50E−03 | branched-chain amino acid catabolic process [GO:0009083] | P | 1.65E−04 |
| carbohydrate biosynthetic process [GO:0016051] | P | 2.50E−03 | leucine catabolic process [GO:0006552] | P | 1.34E−03 |
| pyrimidine nucleoside salvage [GO:0043097] | P | 3.12E−03 | AMP biosynthetic process [GO:0006167] | P | 1.41E−03 |
| O-acetyltransferase activity [GO:0004026] | F | 3.43E−03 | AMP metabolic process [GO:0046033] | P | 1.41E−03 |
| Membrane [GO:0016020] | C | 4.19E−03 | methylated histone binding [GO:0035064] | F | 1.41E−03 |
| uridine kinase activity [GO:0004849] | F | 4.50E−03 | adenylosuccinate synthase activity [GO:0004019] | F | 1.41E−03 |
| nucleoside salvage [GO:0043174] | P | 4.50E−03 | PeBoW complex [GO:0070545] | C | 1.41E−03 |
| plant-type cell wall organization [GO:0009664] | P | 5.70E−03 | cellular amino acid catabolic process [GO:0009063] | P | 1.41E−03 |
| single-organism carbohydrate metabolic process [GO:0005975] | P | 5.83E−03 | phosphoacetylglucosamine mutase activity [GO:0004610] | F | 1.74E−03 |
| Endosome [GO:0005768] | C | 6.68E−03 | de novo' AMP biosynthetic process [GO:0044208] | P | 2.33E−03 |
| cell wall organization [GO:0071555] | P | 8.97E−03 | leucine metabolic process [GO:006551] | P | 3.20E−03 |
| | | | geranylgeranyl-diphosphate geranylgeranyltransferase activity [GO:0016767] | F | 3.26E−03 |
| | | | ELL-EAF complex [GO:0032783] | C | 6.81E−03 |
| | | | regulation of rRNA processing [GO:2000232] | P | 9.23E−03 |

**Notes.**
P, Process; F, Function; C, Cellular.

## DISCUSSION

Previously, there were only two reports that attempted to develop sex identification markers in Asian Palmyra palm. *George et al. (2007)* presented a male-specific OPA-06$_{600}$ marker after screening 180 RAPD primers using ten male and ten female samples from several populations in India and later used this marker to verify the sexes of more than 100 seedlings (*George & Karun, 2011*). *Vinayagam et al. (2009)* reported another attempt, but could not identify any sex-linked marker after screening up to 130 ISSR markers (with 65 polymorphic bands) based on 20 accessions from another population in India. In this work, we have also tested the OPA-06$_{600}$ marker in Thailand populations, which were collected from the central, northeastern and southern regions, but this marker failed to confirm the sexes in our experiment. The limitation of sex-linked markers across varieties and populations is common (*Heikrujam et al., 2015*), and it is most likely due to sequence variation at loci used for developing the markers. Thus, it is required that sex-linked markers for populations in Thailand have to be specifically developed.

The success of identifying molecular markers via DNA fingerprinting may lie upon the genome size and the number of screening PCR primers, and this has been reflected in many reports for developing sex-linked markers (see *Heikrujam et al., 2015* for an extensive review on the numbers of primers used in different dioecious plants). Surprisingly, date palm, which is closely related to Asian Palmyra palm (*Barrett et al., 2016*; *Sakulsathaporn et al., 2017*), readily yielded sex-linked markers when tested with only small numbers of screening primers. For example, *Younis, Ismail & Soliman (2008)* obtained one male-specific and two female-specific markers after screening 30 RAPD primers and five male-specific markers after screening 20 ISSR primers, *Elmeer & Mattat (2012)* identified 22 microsatellite loci for sex-determination in some date palm cultivars after screening 14 SSR primer pairs, and *Dhawan et al. (2013)* identified a male-specific marker after screening 100 RAPD primers. Because the genome size of Asian Palmyra palm has not yet been reported, it is difficult to estimate the number of screening primers to cover the genome for effective screening. Considering that our experiment used an extensive set of 1,204 primer pairs among ten techniques and there was only a single sex-linked marker identified in the previous attempts using up to 180 RAPD and 130 SSR primers (*Vinayagam et al., 2009*; *George & Karun, 2011*), this demonstrated a complexity in identifying sex-linked loci in Asian Palmyra palm.

Sex identification of date palm was subjected to a debate as newly developed sex-linked markers via DNA fingerprinting often failed when tested in other varieties or populations (*Rania & Younis, 2008*; *Al-Mahmoud et al., 2012*; *Elmeer & Mattat, 2012*; *Dhawan et al., 2013*; *Maryam et al., 2016*). Indeed, varieties of date palm germplasm and geographical populations were included as much when developing those markers, but the power of sex identification was limited by the variation of DNA sequences at the sex loci and, at that time, an unclear sex determination system in date palm. This problem has recently been solved by the availability of date palm genomes (*Al-Dous et al., 2011*), verification of sex determination loci (*Cherif et al., 2013*) and construction of genetic map (*Mathew et al., 2014*), validating that date palm has a homomorphic XY chromosome system.

**Table 5  Transcripts differentially expressed between sexes.**

| Transcript IDs | Tested samples | | FPKM S1 | FPKMS2 | Fold change (log2) | *q* value | Blast2GO annotation |
|---|---|---|---|---|---|---|---|
| | S1 | S2 | | | | | |
| c129593_g1_i1[a] | FO | MO | 7.700 | 63.365 | 3.041 | 0.0241 | auxin-induced 15A-like (*Cao et al., 2006*) |
| c42126_g1_i1[a] | FO | MO | 25.601 | 163.217 | 2.673 | 0.0471 | glucan endo-1,3-beta-glucosidase 13 (*Nagai et al., 1999*) |
| c1819_g1_i1[a,b] | FO | MY | 1738.540 | 99.632 | −4.125 | 0.0383 | ethylene-responsive transcription factor 4-like (*Liu et al., 2008*; *Tao et al., 2018*) |
| c142400_g1_i1[a,b] | FO | MY | 949.073 | 68.455 | −3.793 | 0.0087 | ethylene-responsive transcription factor ERF017-like (*Liu et al., 2008*; *Tao et al., 2018*) |
| c128418_g1_i1[a] | FY | MO | 6.541 | 146.348 | 4.484 | 0.0151 | floricaula leafy homolog (*Blázquez et al., 1997*) |
| c119992_g1_i1[a] | FO | MO | 4.834 | 59.838 | 3.630 | 0.0383 | galactinol synthase 1-like (*Fan et al., 2017*) |
| c138805_g1_i2[a] | FO | MY | 4.267 | 34.496 | 3.015 | 0.0491 | argonaute 10 (*Nonomura et al., 2007*) |
| c211381_g1_i1 | FY | MO | 69.478 | 575.104 | 3.049 | 0.0241 | anti-muellerian hormone type-2 receptor [*Arabidopsis thaliana*] |
| c140985_g1_i1 | FY | MY | 130.511 | 20.195 | −2.692 | 0.0409 | ASPARTIC PROTEASE IN GUARD CELL 2-like |
| c195337_g1_i1 | FY | MO | 44.752 | 547.920 | 3.614 | 0.0151 | CASP 2C1 |
| c127228_g1_i1 | FY | MO | 71.119 | 6.415 | −3.471 | 0.0328 | cellulose synthase A catalytic subunit 4 [UDP-forming] |
| c131838_g1_i1 | FY | MY | 20.253 | 2.104 | −3.267 | 0.0289 | cellulose synthase A catalytic subunit 9 [UDP-forming]-like |
| c145948_g1_i1 | FY | MO | 26.670 | 177.451 | 2.734 | 0.0448 | F-box GID2-like |
| c124632_g1_i2 | FY | MO | 3.413 | 43.070 | 3.658 | 0.0383 | high mobility group B 7 [*Nelumbo nucifera* ] |
| c125833_g1_i2 | FO | MY | 181.228 | 30.473 | −2.572 | 0.0460 | ingression fic1-like |
| c104934_g1_i1 | FY | MY | 5.703 | 70.065 | 3.619 | 0.0442 | long-chain-alcohol oxidase FAO4A |
| c138788_g1_i1 | FY | MY | 483.513 | 19.217 | −4.653 | 0.0133 | NDR1 HIN1-Like 3-like |
| c106063_g1_i1 | FO | MY | 458.018 | 47.014 | −3.284 | 0.0198 | nuclease HARBI1 |
| c181538_g1_i1 | FO | MY | 431.202 | 34.566 | −3.641 | 0.0087 | nuclease HARBI1 |
| c134717_g2_i1 | FO | MY | 580.064 | 54.540 | −3.411 | 0.0210 | nuclease HARBI1 |
| c126031_g2_i3 | FO | MY | 142.996 | 13.803 | −3.373 | 0.0210 | nuclease HARBI1 |
| c126031_g2_i2 | FO | MY | 233.676 | 27.069 | −3.110 | 0.0380 | nuclease HARBI1 |
| c116702_g1_i1 | FO | MY | 116.085 | 6.274 | −4.210 | 0.0133 | peroxidase 3-like |
| c129100_g1_i2 | FO | MO | 2.660 | 33.248 | 3.644 | 0.0375 | probable 2-aminoethanethiol dioxygenase |
| c98445_g1_i1[b] | FY | MY | 172.349 | 8.685 | −4.311 | 0.0380 | probable WRKY transcription factor 25 |

**Table 5** (*continued*)

| Transcript IDs | Tested samples | | FPKM S1 | FPKMS2 | Fold change (log2) | *q* value | Blast2GO annotation |
|---|---|---|---|---|---|---|---|
| | S1 | S2 | | | | | |
| c129627_g1_i4[b] | FY | MO | 3.978 | 64.185 | 4.012 | 0.0292 | probable WRKY transcription factor 65 |
| c141291_g1_i12 | FO | MY | 309.255 | 51.579 | −2.584 | 0.0409 | probable xyloglucan endotrans-glucosylase hydrolase 23 |
| c141291_g1_i9 | FO | MY | 141.769 | 24.912 | −2.509 | 0.0500 | probable xyloglucan endotrans-glucosylase hydrolase 23 |
| c128725_g1_i1 | FO | MY | 149.511 | 9.942 | −3.911 | 0.0472 | probable xyloglucan endotrans-glucosylase hydrolase 23 |
| c105494_g1_i1 | FO | MY | 102.967 | 14.931 | −2.786 | 0.0409 | probable xyloglucan endotrans-glucosylase hydrolase 23 |
| c118421_g1_i2 | FY | MY | 54.923 | 2.674 | −4.361 | 0.0241 | remorin [*Eucalyptus grandis*] |
| c138049_g1_i1 | FY | MO | 56.348 | 336.410 | 2.578 | 0.0472 | signal [*Medicago truncatula*] |
| c108722_g1_i1 | FY | MO | 37.435 | 654.878 | 4.129 | 0.0472 | signal [*Medicago truncatula*] |
| c135861_g1_i2 | FY | MO | 38.893 | 3.975 | −3.290 | 0.0383 | trans-resveratrol di-O-methyltransferase-like |
| c126812_g1_i2[b] | FO | MY | 117.456 | 12.564 | −3.225 | 0.0133 | transcription factor bHLH94-like |
| c126812_g1_i1[b] | FO | MY | 71.762 | 5.114 | −3.811 | 0.0241 | transcription factor bHLH94-like |
| c136093_g2_i1[b] | FO | MY | 91.163 | 8.193 | −3.476 | 0.0133 | transcription factor bHLH94-like |
| c155603_g1_i1 | FY | MY | 155.137 | 17.684 | −3.133 | 0.0380 | U-box domain-containing 25-like |
| c82639_g1_i1 | FY | MO | 87.025 | 10.838 | −3.005 | 0.0292 | U-box domain-containing 26-like |
| c139774_g4_i1 | FY | MY | 192.281 | 28.590 | −2.750 | 0.0472 | UPF04964 |
| c135252_g1_i1 | FO | MY | 40.178 | 2.638 | −3.929 | 0.0198 | VQ motif-containing 4-like |
| c137964_g3_i1 | FY | MO | 30.285 | 441.859 | 3.867 | 0.0380 | Ycf68 (chloroplast) [*Carnegiea gigantea*] |
| c137478_g2_i2 | FY | MO | 50.030 | 369.658 | 2.885 | 0.0241 | ycf68 [*Medicago truncatula*] |

**Notes.**
Transcript length > 300 bp, FPKM > 5.
[a] involved in sex determination and flower development.
[b] with high FMKM values.

This sex chromosome is currently being used to develop precise sex-specific markers for specific varieties and populations of date palm (*Mohei et al., 2017*; *Ali et al., 2018*; *Intha & Chaiprasart, 2018*). Thus, it is potential that genome sequencing of Asian Palmyra palm could help us verify the sex determination region and, perhaps, the sex determination system in Asian Palmyra palm.

Potential sex-specific transcripts of Asian Palmyra palm were identified from SSH and *de novo* transcriptome analyses. We have tested many of these sex-specific transcripts for developing sex identification markers by PCR analysis, but none was achieved. Based on a chromosome study by *Sharma & Sarkar (1956)* proposing the XY chromosomes as the sex determination system of Asian Palmyra palm (XY for male and XX for female), it

can be anticipated that transcripts from X chromosome should be detected in both sexes, while transcripts from the male-specific region of the Y chromosome should only be found in the male. Although most male-specific transcripts identified here are expected to be autosomal genes that support anther development or female sterility, some will be encoded by genes on the male specific region of the Y chromosome (MSY). Those male-specific transcripts can be used as markers for sex and are candidates for sex determination genes. However, PCR analysis of male-specific transcripts so far cannot identify any male-specific marker. Studying Y chromosome of Asian Palmyra palm would be essential for further development on male-specific markers.

Through the exhaustive screening of sex-specific markers using the three approaches performed in this work, this may reveal some aspects of the sex chromosome evolution of this palm species. Firstly, it is possible that the Asian Palmyra palm may have evolutionary young sex chromosomes, which recently diverged, and that the MSY is too small to be identified by the scope of this work. It could be interesting to add Asian Palmyra palm as a specimen for studying the evolution of sex chromosomes (*Charlesworth, 2015*). Secondly, although the work by *Sharma & Sarkar (1956)* had depicted clearly large difference of the sex chromosomes in the shape and size, it is still uncertain whether the XY chromosomes is the sex determination system of this palm as it has not been confirmed at molecular levels. Thirdly, because *George et al. (2007)* had identified a male-specific marker, which could not be used in the palm population in Thailand, there may be a unique haplotype of the MSY in Thailand. This haplotype may arise from selected individuals during migration to the southeast Asia (*Pipatchartlearnwong et al., 2017a*) or the sex chromosomes were recently evolved with small sex determination regions that are difficult to detect.

By comparing the SSH and transcriptome data, to our surprise, most SSH clones were uncorrelated to the sexes, given that the cDNA clones were screened through two rounds of dot blot hybridization. This observation may indicate the limitation of the SSH technique for identifying genes in a complex system. Alternatively, this may be because the RNA samples used for the SSH analysis were obtained from different flower stages from those used for the transcriptome sequencing. A potential weakness of the SSH and transcriptome sequencing was that RNA samples were isolated from developing inflorescence tissues, not from inflorescence primordia that initiate sex organs and that key sex determination transcripts may not be present or difficult to identify in the inflorescence stages (*Harkess et al., 2015*). Collecting the primordial tissues for this study is a very challenging task, as the Asian Palmyra palm tip is covered by many layers of thick and hard leaf sheets standing at 20 m height with no indication whether the primordial tissues will develop to be an inflorescence or a leaf, and the plant usually dies after tip removal. Moreover, the variation of sequencing depth among the samples (47–90 million reads per sample) and the lack of replicates could attribute to the complication for extracting robust conclusions from the transcriptome analysis.

Nonetheless, this work provides transcriptome data that would particularly benefit to two research areas: plant sucrose metabolism and sex development in palms and monocots. Sucrose production is one of the important areas in plant biotechnology, and many transcriptome studies have devoted to understand the control of metabolic

flux towards sucrose. Most of the study was performed in major sugar crops including sugarcane (*Cardoso-Silva et al., 2014*; *Huang et al., 2016*), sugar beets (*Mutasa-Göttgens et al., 2012*) and sorghum ((*Mizuno, Kasuga & Kawahigashi, 2016*), and our data add transcript candidates for sucrose metabolism in the inflorescence of palm species. For sex development, male- and female-specific transcripts and differentially expressed transcripts between sexes were listed to provide data for further study in sex determination and male and female inflorescence development. A number of studies with similar objectives in identifying sex determination genes and underlying mechanisms of sex development in dioecious plants via the transcriptome sequencing have been reported in asparagus (*Harkess et al., 2015*; *Li et al., 2017*), *Idesia polycarpa* (*Mei et al., 2017*), shrub willows (*Salix suchowensis*) (*Liu et al., 2013*), poplar (*Song et al., 2013*) and *Coccinia grandis* (*Mohanty et al., 2017*). Most of the identified genes belong to floral development, phytohormone biosynthesis, hormone signaling and transduction, transcriptional regulation and DNA methyltransferase activity. However, with many genes playing the role during the complex developmental process, it is difficult to determine the mechanism underlying the sexual development and sex determination in dioecious species. Future progress in functional genomics addressing the identified genes would be an essential tool to solve this long-standing question.

## CONCLUSIONS

Although no sex-linked marker was obtained from exhaustive DNA fingerprinting, SSH and *de novo* transcriptome analysis, this work provides transcripts based on male and female inflorescences of the Asian Palmyra palm. Further attempts on developing sex identification markers in Asian Palmyra palm should be directed towards genomic-based approaches, particularly at the MSY. Genome analysis using SNPs have been successful in accessing sex determination loci in date palm (*Al-Mahmoud et al., 2012*; *Ali et al., 2018*) and other dioecious plant species (*Zhou et al., 2018*; *Jia et al., 2019*). Whole genome sequencing, genetic mapping, SNPs and genome-wide association study (GWAS) between male and female populations would be essential tools for further identification of sex-linked loci in Asian Palmyra palm.

### Funding

This work was financially supported by Kasetsart University Research and Development Institute (KURDI), Faculty of Science Research Fund (ScRF) and OmiKU. Supachai Vuttipongchaikij was supported by Thailand Research Fund (TRF-RSA6080031). Kwanjai Pipatchartlearnwong was supported by the PhD studentship from Kasetsart University and Faculty of Science. The funders had no role in study design, data collection and analysis, decision to publish, or preparation of the manuscript.

### Grant Disclosures

The following grant information was disclosed by the authors:

Kasetsart University Research and Development Institute (KURDI).
Faculty of Science Research Fund (ScRF).
Thailand Research Fund: TRF-RSA6080031.
Kasetsart University and Faculty of Science.

## Competing Interests

The authors declare there are no competing interests.

## Author Contributions

- Kwanjai Pipatchartlearnwong performed the experiments, analyzed the data, prepared figures and/or tables, authored or reviewed drafts of the paper, approved the final draft.
- Piyada Juntawong and Passorn Wonnapinij performed the experiments, analyzed the data, contributed reagents/materials/analysis tools, prepared figures and/or tables, approved the final draft.
- Somsak Apisitwanich conceived and designed the experiments, contributed reagents/materials/analysis tools, authored or reviewed drafts of the paper, approved the final draft.
- Supachai Vuttipongchaikij conceived and designed the experiments, analyzed the data, contributed reagents/materials/analysis tools, prepared figures and/or tables, authored or reviewed drafts of the paper, approved the final draft.

## Field Study Permissions

The following information was supplied relating to field study approvals (i.e., approving body and any reference numbers):

Sample collections were conducted on privately-owned lands.

## Data Availability

The sequence of SSH clones are available in GenBank (JZ977504–JZ977592). The Transcriptome Shotgun Assembly are available at DDBJ/ENA/GenBank (accession GFYQ00000000). Other raw data are available in Figs. S1–S4 and Tables S1–S15.

## Supplemental Information

Supplemental information for this article can be found online at http://dx.doi.org/10.7717/peerj.7268#supplemental-information.

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
