# Peer review of "Towards sex identification of Asian Palmyra palm (Borassus flabellifer L.) by DNA fingerprinting, suppression subtractive hybridization and de novo transcriptome sequencing"

_PeerJ, doi:10.7717/peerj.7268_

## Round 0.1 · original submission · Minor Revisions

Thank you for submitting this valuable contribution towards identification of sex markers in palms. Both the reviewers and I agree that only minor changes are needed to ready this manuscript for publication. Please make the requested changes.

Reviewer 1 ·

Basic reporting

The manuscript “Towards sex identification of Asian Palmyra palm (Borassus flabellifer L.) by DNA fingerprinting, suppression subtractive hybridization and de novo transcriptome sequencing” attempts to identify sex-linked markers for this palm using PCR-based DNA fingerprinting, suppression subtractive hybridization (SSH) and transcriptome sequencing. Overall, the manuscript conforms to the structure recommended by the journal viz. Introduction, Materials and Methods, Results, Discussion, and Conclusions. The findings of the study are well discussed and supported by existing evidence. The overall English comprehension and grammar of the article is good and attains standards of a scientific article. I really appreciate that authors have well-presented the negative results and discussed the underlying complexity in identifying sex-related markers in Asian Palmyra palm. However, below are a few suggestions that would be useful for the readers and make it scientifically thorough.

1. In line 123, “using the followed condition should be replaced by “using the conditions as follow:”
2. Please use either “version” or “ ver” consistently throughout the manuscript to refer to software versions.
3. In line 193-195, the sentence is ambiguous/incomplete. Please rephrase/update it. Maybe break it in two sentences.
4. Please replace Go with GO throughout the text, wherever applicable.
5. In line 423, “understanding” should be replaced by “understand”.

Experimental design

1. The two statements from line 178 to 182 are contradictory with respect to quality score cutoff used for filtering. Please be consistent if the author set the quality score cutoff of 20 or 25 to filter low quality reads.
2. In line 184-186 for similarity search, the percentage of alignment length matched and %identity are also useful parameter to consider along with E-value. Can the author mention in the text avg? alignment length (%) matched and average %identity exclusively in the text along with E-value cutoff?
3. In line 190-191, please mention specifically in the text which 4 transcriptomes are used for combined assembly using Cufflinks? For e.g. two floral stages of male and female (FY, FO, MY, and MO).
4. In the “Methods” section, please include the steps/criteria to perform GO enrichment analysis of for the male- and female-specific datasets.
5. In the “Methods” section, please include the steps/criteria to identify a) transcripts that were shared among the datasets b) specific to one transcriptome).

Validity of the findings

The study is experimentally sound and well presented. However, the findings of the transcriptome analysis would have been more robust with the use of replicates per sample. Moreover, huge variation (47,194,682-90,305,610) in the sequencing depth of the samples makes it more complicated to extract robust conclusions. For further comments, refer section Experimental design.

Reviewer 2 ·

Basic reporting

no comment

Experimental design

no comment

Validity of the findings

no comment

Additional comments

I would recommend the following minor revisions:

1) Line 67: Replace “is” by “are” in the first sentence.

2) Lines 75-76: I would suggest to delete this sentence because it does not make any sense.

3) Line 110: Replace “centrifugation” by “centrifuged” in the first sentence.

4) Lines 175-176: Total RNA was obtained NOT from one male and one female inflorescence samples, but from two male (MY and MO) and two female (FY and FO) inflorescence samples. Please, correct it.

5) Lines 194-195: It is unclear what you meant by “…, unless two transcripts have corresponded of four transcriptomes”. Please, rephrase it.

6) Line 245: Provide also full names for abbreviations FY, FO, MY and MO, such as, FY - female young inflorescences, FO - female old inflorescences, etc.

7) Lines 324-325: Please, list those genes for seven annotated transcripts that were previously identified to be involved in sex determination and flower development, and provide the references for them.

8) Line 395: Replace “potentially” by “possible”, and “young” by “evolutionary young”.

9) Line 399: Replace “… depicted clearly regarding the large difference of the shape and size of the sex chromosomes, …” by ““… depicted clearly large difference of the sex chromosomes in the shape and size, …”

10) Line 429: Replace “study” by “studies”.

11) Line 431: Replace “has been” by “have been”.

11) Line 447: Cite also Ali et al., 2018 together with Al-Mahmoud et al., 2012.

Best regards,

---

## Round 0.2 · Minor Revisions

This is basically a failed experiment for which another analysis was embedded to help save the efforts offered. The annotation for the GO terms was mentioned in Tables 4 and 5, but the actual GO number is missing; it would be great if these numbers could be additionally attached (e.g. GO:123456). Are the sequences for the Transcript_IDs listed in Table 5 available by the same names in the sequence repository? This is suggested within the declarations section. If not, these should be placed as additional supplemental data. In general this is a well presented manuscript, and explains it shortcomings sufficiently. I am generally inclined to ACCEPT this manuscript if the GO: annotations and sequence connections identified in the manuscript can be addressed.

Reviewer 2 ·

Basic reporting

The authors fully complied with my requests.

Experimental design

sufficient

Validity of the findings

sufficient

Additional comments

No additional comments, but I would recommend additional revisions for Table 5 in the file attached.

Annotated reviews are not available for download in order to protect the identity of reviewers who chose to remain anonymous.

---

## Round 0.3 · accepted · Accept

Thank you for the additional edits regarding the addition of GO terms to the identified gene sets. This may be of value for those that attempt to re-visit this research question. When answers appear we sometimes discover that the answer was always there. Best wishes in your further research. Please consider this manuscript accepted.